# Growth Differentiation Factor 15: A Biomarker with High Clinical Potential in the Evaluation of Kidney Transplant Candidates

**DOI:** 10.3390/jcm9124112

**Published:** 2020-12-20

**Authors:** Marina de Cos Gomez, Adalberto Benito Hernandez, Maria Teresa Garcia Unzueta, Jaime Mazon Ruiz, Covadonga Lopez del Moral Cuesta, Jose Luis Perez Canga, David San Segundo Arribas, Rosalia Valero San Cecilio, Juan Carlos Ruiz San Millan, Emilio Rodrigo Calabia

**Affiliations:** 1Nephrology Department, Hospital Universitario Marques de Valdecilla, Avenida Valdecilla n 5, 39724 Santander, Spain; abenito@humv.es (A.B.H.); jaime.mazon@scsalud.es (J.M.R.); covalk@hotmail.com (C.L.d.M.C.); jlperezcanga@gmail.com (J.L.P.C.); rosalia.valero@scsalud.es (R.V.S.C.); juancarlos.ruiz@scsalud.es (J.C.R.S.M.); emilio.rodrigo@scsalud.es (E.R.C.); 2Valdecilla Biomedical Research Institute (IDIVAL), Cardenal Herrera Oria S/N, 39011 Santander, Spain; mteresa.garciau@scsalud.es (M.T.G.U.); david.sansegundo@scsalud.es (D.S.S.A.); 3Clinical Analysis Department, Hospital Universitario Marques de Valdecilla, Avenida Valdecilla n 5, 39724 Santander, Spain; 4Clinical Immunology Department, Hospital Universitario Marques de Valdecilla, Avenida Valdecilla n 5, 39724 Santander, Spain

**Keywords:** early mortality, growth differentiation factor 15, kidney transplantation, pretransplant assessment, survival

## Abstract

Kidney transplantation implies a significant improvement in patient survival. Nevertheless, early mortality after transplant remains high. Growth differentiation factor 15 (GDF-15) is a novel biomarker under study as a mortality predictor in multiple scenarios. The aim of this study is to assess the utility of GDF-15 to predict survival in kidney transplant candidates. For this purpose, 395 kidney transplant recipients with pretransplant stored serum samples were included. The median GDF-15 was 5331.3 (50.49–16242.3) pg/mL. After a mean of 90.6 ± 41.5 months of follow-up, 82 (20.8%) patients died. Patients with higher GDF-15 levels (high risk tertile) had a doubled risk of mortality after adjustment by clinical characteristics (*p* = 0.009). After adjustment by EPTS (Estimated Post Transplant Survival score) the association remained significant for medium hazards ratios (HR) 3.24 95%CI (1.2–8.8), *p* = 0.021 and high risk tertiles HR 4.3 95%CI (1.65–11.54), *p* = 0.003. GDF-15 improved the prognostic accuracy of EPTS at 1-year (ΔAUC = 0.09, *p* = 0.039) and 3-year mortality (ΔAUC = 0.11, *p* = 0.036). Our study suggests an independent association between higher GDF-15 levels and mortality after kidney transplant, adding accuracy to the EPTS score, an established risk prediction model currently used in kidney transplant candidates.

## 1. Introduction

Growth differentiation factor 15 (GDF-15) is a stress-responsive member of transforming growth factor β (TGF-β) cytokine superfamily. This pleiotropic protein has been linked to inflammation, metabolism, and oncogenesis regulation. Poorly expressed in healthy individuals, this molecule is upregulated in many pathological conditions such as following injury, ischemia, and other forms of oxidative and/or metabolic stress, raising interest in its potential utility as a biomarker in human disorders [1,2,3]. 

Several studies have demonstrated the connection between increased circulating GDF-15 concentrations and the development of cardiovascular pathologies. In patients with heart failure, GDF-15 has added value beyond the traditional risk factors that predict all-cause and vascular mortality [3,4]. Increased levels of GDF-15 are seen in patients with lower ejection fraction, diastolic dysfunction, and more severe symptoms of heart failure [5,6]. GDF-15 strongly increases after myocardial infarction. Compared to necrosis biomarkers, its elevation is not related to the extent of myocardial damage, but to chronic disease burden and worse outcomes. In these patients, higher levels have been associated with increased risk of mortality and recurrent myocardial infarction. Due to this ability, it has been proposed as a biomarker for stratification and decision management in the setting of acute coronary syndrome [7,8,9]. Outside the heart, GDF-15 has also been related to other vascular pathologies, including peripheral vascular disease [10] and stroke [11,12]. 

Although not so extensively studied, GDF-15 may be related to cancer diagnosis and tumor progression. Its relation with angiogenesis, proliferation, apoptosis, and growth remodeling support this idea. Elevated expression of GDF-15 has been demonstrated in different types of cancer including breast, colorectal, prostate, and head and neck, among others [13,14,15,16]. 

Regarding nephrology, increased GDF-15 levels have been associated with higher risk of incident chronic kidney disease and more rapid decline in kidney function in different renal disorders, including diabetic nephropathy, IgA nephropathy, and primary membranous nephropathy [17,18,19,20]. Over the last years, some studies have focused in the prognostic utility of GDF-15 in chronic kidney disease population. Equally to the general population, higher circulating serum GDF-15 concentrations have been related with mortality and vascular events in all CKD stages [21,22,23,24], including hemodialysis patients [25,26]. GDF-15 is generally increased in patients with kidney disease and associated with dialysis vintage in hemodialysis patients. Nevertheless, little data has been published regarding GDF-15 in kidney transplant recipients. The levels of this biomarker seem to highly decrease after transplantation compared to dialysis, in line with the reduction of cardiovascular risk and mortality [27,28,29]. GDF-15 has also been related to anemia in kidney transplant recipients [30] and the degree of ischemia reperfusion injury after transplant [31]. 

Cardiovascular disease is the most common cause of death in transplant recipients, followed by infections and cancer [32,33]. The risk of major adverse cardiac events (MACE) rises in the early post-transplant period and declines to a lower rate thereafter compared with dialysis patients [34,35]. In spite of that, cardiovascular morbi-mortality remains highly prevalent in this population, and assessing cardiovascular (CV) status is key in every pretransplant evaluation [36,37]. However, current methods do not seem accurate enough to strongly prevent these events [38,39,40]. In this sense, it would be extremely useful to develop non-invasive and user-friendly biomarkers to better stratify death risk and other post-transplant complications, avoiding inconclusive aggressive tests. Given the relation of GDF-15 with many of the pathogenic conditions presented in kidney transplant recipients, this could be an ideal marker to study pretransplant. Hence, the aim of this study is to define the determinants of GDF-15 in a cohort of advanced chronic kidney disease patients and evaluate its potential use as a predictor of survival after transplant.

## 2. Experimental Section

We analyzed all adult renal transplants performed in our center from 2005 to 2015. As routine clinical practice in our hospital since 1985, serum samples were collected pretransplant and cold stored. Only pretransplant samples with at least 150 microliters available for GDF-15 testing were included in our study. The study was conducted according to the guidelines of the Declaration of Helsinki and approved by the hospital’s Ethics Committee.

The concentration of GDF-15 was analyzed using an enzyme-linked immunosorbent assay (Quantikine, R&D Systems, Minneapolis, MN, USA). All samples with an elevated level were diluted to provide quantitative results. We performed duplicate measurements for each sample with good repeatability. In this study, GDF-15 was analyzed as a continuous variable after log transformation (as it was not normally distributed) and as categorical, dividing our cohort in GDF-15 tertiles. 

Relevant information about recipient, donor, and transplant characteristics was extracted from the prospectively maintained database of renal transplant patients in our center. Clinical characteristics obtained from the medical records included age, sex, race, cause of CKD, dialysis, number of transplants, and history of other solid organ transplants, among others. Regarding CV events, we included coronary artery disease (based on imaging confirmation) and clinically significant (requiring radiological or surgical intervention) peripheral arteriopathy. Diagnosis of pretransplant diabetes was considered if the patient required hypoglycemic treatment prior transplant.

Estimated Post Transplant Survival (EPTS) score was calculated in all first renal transplant recipients included in our study. The factors needed in the formula (age in years, time on dialysis in years, diagnosis of diabetes, and history of previous solid organ transplant) were extracted from our database and calculated in the official webpage: https://optn.transplant.hrsa.gov/resources/allocation-calculators/epts-calculator/.

Continuous variables were expressed as mean ± standard deviation or median with interquartile range (IQR) if not normally distributed. Categorical variables were described as relative frequencies. Univariate and multivariate logistic regression was performed to determine which clinical variables were associated with higher levels of GDF-15 in our population.

The primary outcome of our study was all-cause mortality. Cox proportional hazards regression was used to test the association of GDF-15 (log GDF-15 and tertiles) with mortality. Covariates for adjustment were selected a priori based on previous literature. Statistically significant variables related to mortality in the univariate model were included in the multivariate Cox regression analysis. Hazards ratios (HR) were reported with 95% confidence intervals (CI 95%). A *p* value of less than 5% was reported as statistically significant. Receiver operating characteristic (ROC) curves and area under the curve (AUC) were obtained to compare the predicted probability of logistic regression models using GDF-15, EPTS, and the combination of both to predict mortality 1, 3, and 5 years post-transplant. For our analysis, we used SPSS, version 22.0 (SPSS Inc., Chicago, IL, USA) and MedCalc^®^ Statistical Software version 19.6 (MedCalc Software Ltd., Ostend, Belgium).

## 3. Results

Four hundred and fifty patients were transplanted in our center from January 2005 to December 2015 (both included). Of them, 395 had stored serum samples obtained immediately pretransplant and were therefore included in our study. The samples were obtained after the short vintage in hemodialysis patients and immediately pretransplant in peritoneal dialysis and preemptive transplant candidates. Baseline characteristics of the study population are shown in Table 1. Thirty-one patients (7.8%) were non-kidney transplant recipients: 5 liver, 8 heart, 4 pulmonary, and 14 pancreas receptors. Two hundred and sixty-four patients (66.8%) received their first renal transplant during the study period, whereas 131 were retransplants. The total time of renal replacement therapy (including dialysis and former transplants) was 1.76 (IQR 0.56–5.06) years. Three hundred and nineteen recipients were not previously sensitized (virtual panel-reactive antibody (PRA) 0%). Among the sensitized ones (*n* = 76), the median peak PRA was 65% (IQR 36.7–93.3). The immunosuppression was based on anticalcineurin inhibitors (91.4%), mTOR inhibitors (4.6%), mycophenolate or azathioprine (93.9%), and prednisone (93.4%). Two hundred and seventy-five patients (69.6%) did not receive induction therapy, whereas 18 (4.6%) received basiliximab, and 100 (25.4%) received thymoglobulin. Regarding donor type, 5.8% of patients received a living donor transplant, while 94.2% were from deceased donors (92.4% brain death donors and 1.8% cardiac death donors).

The median GDF-15 in our cohort was 5331.29 (IQR 4071.83–6819.93) pg/mL. Only 6 patients (1.5%) had an GDF-15 level ≤2000 pg/mL (normal level for healthy reference population) [1,15]. Patients were stratified in tertiles according to GDF-15 levels: Low (GDF-15 < 4612.09 pg/mL), medium (GDF-15 4612.09–6296.47 pg/mL), and high risk tertile (GDF-15 > 6296.47 pg/mL). The following variables were associated with higher levels of GDF-15 (>4612.09 pg/mL, medium and high risk tertiles) by univariate logistic regression analysis: Age (OR 1.04 CI95% (1.02–1.06), *p* < 0.001), coronary artery disease (OR 2.66 CI95% (1.14–6.16), *p* = 0.02), peripheral artery disease (OR 2.51 CI95% (1.01–6.2), *p* = 0.048), and time of renal replacement therapy (RRT) (OR 1.04 CI95% (1.01–1.1), *p* = 0.02). GDF-15 was not associated with sex, race, diabetes, primary renal diagnosis, hemoglobin, albumin, creatinine, uric acid, C-reactive protein, phosphorus, parathyroid hormone, preemptive transplant, dialysis modality, or history of non-renal transplants. On multivariate analysis, only two variables were independently related to GDF-15 levels: Age (OR 1.04 (1.02–1.06) *p* < 0.001) and time of RRT (OR 1.04 (1.01–1.07) *p* = 0.02). 

The mean follow-up post-transplant was 90.6 ± 41.52 months. There was no loss to follow-up during the course of the study. During this period, 82 (20.8%) patients died. By univariate Cox analysis, higher GDF-15 concentrations were significantly associated with mortality (log GDF-15 level HR 3.88 CI95% (1.18–12.78), *p* = 0.026). Survival was significantly decreased among the medium and high risk tertiles of GDF-15 compared to the low risk one: HR 2.16 CI95% (1.14–1.44), *p* = 0.018 for medium tertile and HR 3.28 CI95% (1.79–6.1), *p* < 0.001 for high risk tertile (Figure 1). The following variables were also related to mortality by univariate Cox analysis: Age in years (HR 1.07 CI95% (1.04–1.09), *p* < 0.001), diabetes (HR 2.34 CI95% (1.49–3.67), *p* < 0.001), coronary artery disease (HR 2.99 CI95% (1.75–5.13), *p* < 0.001), peripheral arteriopathy (HR 2.24 CI95% (1.24–4.06), *p* = 0.008), other organ solid transplants (HR 1.97 CI95% (1.02–3.84), *p* = 0.044), and renal graft loss (HR 1.61 CI95% (1.03–2.53), *p* = 0.038). Additional variables considered in these analyses, but not significantly related to survival, included: Sex, race, time of RRT and modality, use of dialysis at the time of transplant, albumin, creatinine, uric acid, C-reactive protein, phosphorus, parathyroid hormone, hemoglobin, PRA (%), donor type, use of induction, and type of immunosuppression therapy. 

By multivariate Cox analysis, the relation between survival and GDF-15 remained significant for high risk tertile (HR 2.29 CI95% (1.24–4.24), *p* = 0.009) after adjusting by age (HR 1.07 CI95% (1.04–1.09), *p* < 0.001), history of coronary artery disease (HR 2.2 CI95% (1.26–3.82), *p* = 0.005), censored graft loss (HR 1.95 CI95% (1.23–3.09), *p* = 0.005), and other organ solid transplants (HR 2.64 CI95% (1.32–5.28), *p* = 0.006). Variables associated with mortality by univariate and multivariate Cox regression analysis in our study are summarized in Table 2. 

EPTS score was calculated for all first kidney transplant recipients included in our study (*n* = 264). The EPTS value in our cohort was 31% (IQR 11.5–50.5). After including EPTS in the multivariate analysis, the association between GDF-15 and mortality remained significant. This relation was significant both considering GDF-15 as a continuous variable (logGDF-15 HR 9.46 CI95% (1.78–40.41), *p* = 0.008), or in tertiles (HR 3.24 CI95% (1.2–8.8), *p* 0.021 for medium risk tertile and HR 4.3 CI95% (1.65–11.54), *p* 0.003 for high risk tertile). 

Among the first kidney transplant recipients, mortality at 1, 3, and 5 years was 3% (*n* = 8), 5.7% (*n* = 15), and 7.2% (*n* = 19). By multivariate logistic regression, both EPTS score and high risk GDF-15 tertile were significantly associated with mortality after 1 and 3 years, while mortality at 5 years was only related with EPTS in our analysis. ROC curves based on predicted probability of logistic regression were obtained and compared, showing improved mortality prediction at 1 and 3 years using EPTS combined with GDF-15 tertile (Figure 2).

With regard to the causes of death, cardiovascular and cancer-related mortality were specifically evaluated. Cardiovascular mortality was responsible for 35.4% of deaths (*n* = 29). Malignancy and its treatment complications were responsible for 22.0% of deaths (*n* = 18). By univariate Cox analysis, higher GDF-15 concentrations were significantly associated with cardiovascular mortality: HR 5.95 CI95% (1.32–26.88), *p* = 0.02 for medium tertile, and HR 7.91 CI95% (1.78–35.10), *p* = 0.006 for high risk tertile. Furthermore, this association remained significant (highest GDF-15 tertile HR 5.55 CI95% (1.24–24.7), *p* = 0.025) after adjusting by clinical characteristics associated with cardiovascular mortality in our study: History of coronary artery disease (HR 3.6 CI95% (1.52–8.43), *p* = 0.005), age (HR 1.06 CI95% (1.02–1.11), *p* = 0.007), and diabetes (HR 2.37 CI95% (1.09–7.15), *p* = 0.03). On the contrary, cancer-related mortality was not significantly related to GDF-15 in our analysis: Malignancy related mortality for high risk tertile HR 0.54 CI95% (0.16–1.76), *p* = 0.32.

## 4. Discussion

Death risk prediction after renal transplant remains challenging [36]. In this retrospective cohort study of 395 kidney transplant recipients, we demonstrated the independent association between pre-transplant GDF-15 and post-transplant mortality. This relation remained significant after the adjustment by clinical characteristics and laboratory findings associated with this adverse outcome in previous studies [41,42,43,44]. The highest tertile of GDF-15 in our cohort was associated with more than twice the mortality risk after adjustment. Additionally, GDF-15 improved prognostic yield of EPTS score, a tool that is currently used in some regions to predict mortality and optimize organ allocation [45]. The higher predicting value of EPTS + GDF-15 compared to EPTS alone was observed in the first years after transplant, when better prognostic efficacy is needed. This finding raises the interest in the potential use of this marker in the evaluation of kidney transplant candidates. 

As previously described, the prognostic value of GDF-15 has been studied in general population and other groups of patients [1]. Despite the previous studies evaluating GDF-15 in kidney transplant recipients, the study is the first to our knowledge to investigate its prognostic utility in kidney transplant candidates. The biggest study published to date with regard to the prognostic utility of GDF-15 in CKD patients demonstrated a doubled mortality risk per 1-SD greater after adjustment, that correlates with the findings in our study [24]. More interestingly, our study demonstrates that GDF-15 prediction value remains useful after transplant, even considering the survival improvement seen after this procedure. 

Two different groups have demonstrated the strong relation of this biomarker with mortality in hemodialysis patients, showing better prediction of early (within the first 3 years) adverse events [25,46]. This supports the findings of our study, in which GDF-15 selectively improved the prognostic yield during the first years post-transplant. Recently, Connelly et al. and Jehn et al. [27,29] published a significant fall of GDF-15 after kidney transplantation. This reduction correlated with improved kidney function and reduced myocardial stretch (indicated by NT-proBNP) after transplantation. 

Additionally, we analyzed the causes of mortality in our patients and their relation with GDF-15. As it has been previously described, this biomarker is strongly related with vascular disorders, which remain the main cause of death after a kidney transplants [32,33]. In our cohort, GDF-15 related to cardiovascular mortality even after the adjustment by clinical characteristics that traditionally predict cardiovascular events. On the contrary, our study failed to establish a relation between GDF-15 and cancer-related mortality, in spite of the fact that this relation has been described in the literature. This could be caused by a lack of statistical power due to a relatively small number of events (*n* = 18). It could also be possible that in chronic kidney disease patients, other conditions such as cardiovascular burden may have a greater influence in GDF-15 levels than carcinogenic factors.

In healthy subjects, an increase in serum GDF-15 concentrations is observed with aging (the main determinant factor of GDF-15 in majority of studies) and smoking. A variable relation has been seen among GDF-15 and body mass index, with increased levels in both extremely high and low body mass index [1]. In other cohorts of CKD patients, higher concentrations of GDF-15 were associated with female sex, older age, active smoking, and prevalent diabetes [24]. Other studies have shown its relation with glomerular filtration rate, suggesting that GDF-15 may be a surrogate marker of kidney function. In our study, higher GDF-15 levels were seen in older patients with longer time in RRT, but no other clinical characteristics (such as sex or diabetes) or laboratory findings were associated with its value. These results are similar to the studies developed in dialysis patients, in which age, dialysis duration and markers of nutrition were associated with GDF-15 over other baseline clinical characteristics, indicating the substantial weight of renal history in the pathogenic pathways altered in this group, leading to cardiovascular events and death [25,26]. However, in our study there was not an independent relation between the levels of the marker and RRT modality, in comparison to the results of Jehn et al. [29], therefore more studies should be performed to clarify this point. 

In addition to this, there is still a number of unanswered questions regarding the pathogenesis of GDF-15. As mentioned before, this protein is poorly expressed in healthy tissues, and its production is induced after different forms of cellular damage. However, the source of production in patients with chronic kidney disease remains unknown, as most works have focused on its biomarker value and not on the role as a biological response modifier. Nair et al. [19] demonstrated the expression of GDF-15 in kidney tissue and its correlation with circulating GDF-15 levels, as well as its association with increased risk of CKD progression. In our cohort, GDF-15 did not correlate to renal history (primary disease, preemptive transplant, or RRT modality). As all the patients in our study suffered from advanced stages of kidney disease, we could hypothesize that the differences seen in GDF-15 levels come from the expression in other locations, such as myocardium or vascular tissue. For this reason, more studies are warranted to elucidate the molecular pathways involving GDF-15 and to see whether it plays a prognostic or also a causal role for mortality. This condition would be necessary to translate our findings into clinical practice. 

A large number of mortality markers after kidney transplant have been studied so far. Despite this, their utility remains unclear, as discussed previously [37]. Given the potential benefits of screening this high-risk population, the evolution of this field seems key to improve transplantation results. Imaging studies, such as echocardiogram or cardiac CT scan, are widespread before renal transplantation, even though their utility to reduce cardiovascular burden and mortality after transplant has not been clearly set [36,47]. The use of laboratory tests, in combination with these studies, could improve its prognostic value. They could also be used to limit imaging studies to the highest risk patients and to add objective measures to these studies, otherwise subjected to variability due to the observer’s interpretation. Regarding laboratory biomarkers, some have demonstrated promising results, such us cardiac troponin and soluble ST2. However, they need to prove an additional value over clinical standardized risk models [48,49,50,51]. In our study, GDF-15 achieved this objective, probably because of its pleiotropic nature that affects many pathogenic pathways altered in kidney transplant candidates. Besides that, a fully automated electrochemiluminescence immunoassay for GDF-15 has been presented recently, showing robust analytical performance under clinical practice conditions, which could make its measurement easily available and eventually more cost-effective [52]. 

The strengths of our study include the following; first, a relatively large number of patients with a uniform collection of samples to analyze GDF-15 with measurements performed in a single laboratory. Second, the use of a prospectively maintained database of renal transplant patients, with a follow-up of at least five years after transplant in the same center and no loss of follow-up during the study. Third, the GDF-15 evaluation added prognostic value to a previously widely validated tool for kidney transplant candidates, the EPTS score. 

The limitations of our study also need to be considered. Our population is very homogenous, with a majority of recipients of Caucasian race and grafts obtained mainly from donors after brain death, restricting our external validation. Additionally, we could not compare the use of GDF-15 against other markers such as imaging (echocardiography, intima-media thickness) or anthropometric studies (ankle–brachial index, pulse wave velocity) as they are not routinely performed in all kidney transplant candidates in our center. Furthermore, GDF-15 concentrations were measured only once in each patient, so the evolution of the biomarker concentrations after transplant and the prognostic value of its change could not be assessed. 

In conclusion, our study supports an independent association between higher GDF-15 levels and mortality after kidney transplant, adding accuracy of EPTS score, a stablished risk prediction model currently used in kidney transplant candidates. As an emerging biomarker, further studies are needed to confirm our findings and improve the knowledge regarding this protein, including its exact mechanism of action and potential therapeutic targets, so this knowledge could lead to better outcomes in this population. 

## Figures and Tables

**Figure 1 jcm-09-04112-f001:**
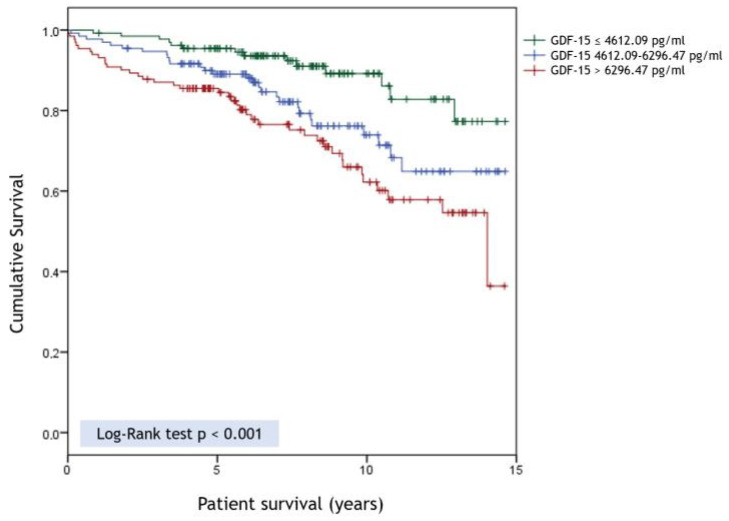
Unadjusted Kaplan–Meier estimates of patient survival according to growth differentiation factor 15 (GDF-15) tertiles.

**Figure 2 jcm-09-04112-f002:**
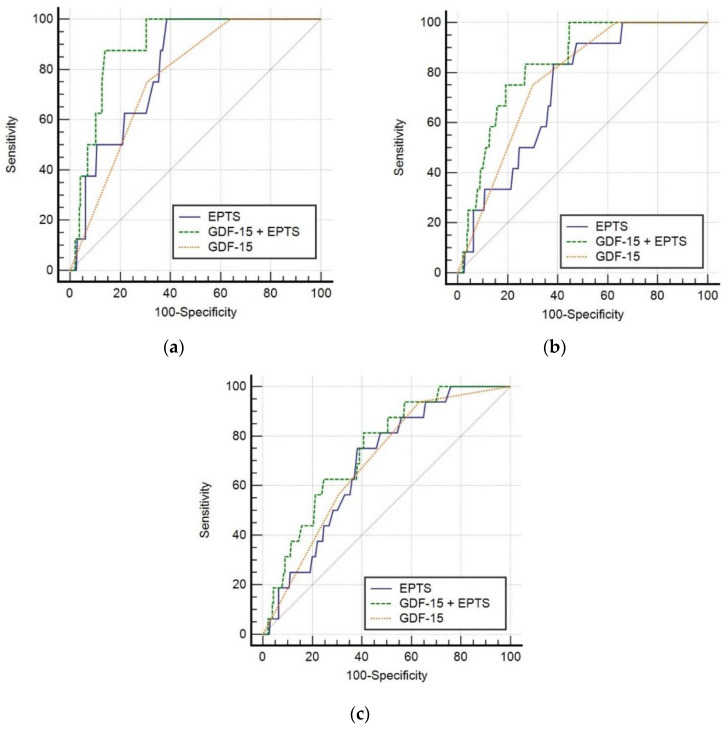
ROC curves based on predicted probability of logistic regression with EPTS, GDF-15 tertiles and EPTS + GDF-15 tertiles. (**a**) 1-year mortality prediction: AUC EPTS 0.81 CI95% (0.67–0.89) vs. AUC EPTS + GDF-15 0.90 CI95% (0.81–0.94), ΔAUC = 0.09 (*p* = 0.039), AUC EPTS vs. AUC GDF-15 0.766 CI95% (0.71–0.81), ΔAUC = 0.04 (*p* = 0.55). (**b**) 3-year mortality prediction: AUC EPTS 0.73 CI95% (0.59–0.82) vs. AUC EPTS + GDF-15 0.83 CI95% (0.71–0.90), ΔAUC = 0.11 (*p* = 0.036), AUC EPTS vs. AUC GDF-15 0.77 CI95% (0.71–0.82), ΔAUC = 0.04 (*p* = 0.56). (**c**) 5-year mortality prediction: AUC EPTS 0.69 CI95% (0.56–0.78) vs. AUC EPTS + GDF-15 0.74 CI95% (0.61–0.83), ΔAUC = 0.06 (*p* = 0.22), AUC EPTS vs. AUC GDF-15 0.69 CI95% (0.62–0.74), ΔAUC = 0.002 (*p* = 0.98). EPTS: estimated post transplant survival; GDF-15: growth differentiation factor 15.

**Table 1 jcm-09-04112-t001:** Baseline characteristics.

**Number of patients**	395
**Donor age (years)**	57.1 (47.6–66.6)
**Cold Ischemia Time (min)**	19 (15–23)
**Recipient age (years)**	52 ± 12.4
**Recipient sex (%)**	68.1 males/31.9 females
**Recipient race (% Caucasian)**	96.5
**Diabetes (%)**	23
**Type I**	7.8
**Type II**	15.2
**Coronary artery disease (%)**	10.4
**Peripheral vascular disease (%)**	8.6
**Non renal solid organ transplant (%)**	7.8
**Primary renal diagnosis**	
**Glomerular (%)**	27.4
**Diabetes (%)**	14.7
**Hypertension/vascular (%)**	24.2
**Polycystic kidney disease (%)**	12.4
**Other (%)**	15.7
**Unknown (%)**	5.6
**Preemptive transplant (%)**	15.9
**Time of renal replacement therapy (years)**	1.76 (0–5.1)
**Retransplant (%)**	33.2
**GDF-15 (pg/mL)**	5331.3 (4071.8–6819.9)
**Hemoglobin (g/dL)**	11.9 (10.8–13)
**Serum albumin (g/dL)**	4 (3.8–4.3)
**Creatinine (mg/dL)**	6.4 (4.9–8.3)
**Uric acid (mg/dL)**	6.3 (5.2–7.8)
**C-reactive protein (mg/L)**	0.5 (0.2–1.1)
**Phosphorus (md/dL)**	5.1 (4.0–6.1)
**Parathyroid hormone (pg/mL)**	290 (149–495)

**Table 2 jcm-09-04112-t002:** Variables associated with mortality in univariate and multivariate Cox regression analysis.

	Univariate		Multivariate			
	HR (CI)	*p*-Value	HR (CI) Model 1	*p*-Value	HR (CI) Model 2	*p*-Value
Age (per year)	1.07 (1.04–1.09)	<0.001	1.07 (1.04–1.09)	<0.001		
Diabetes	2.34 (1.49–3.67)	<0.001		ns		
Coronary artery disease	2.99 (1.75–5.13)	<0.001	2.2 (1.26–3.82)	0.005		
Peripheral arteriopathy	2.24 (1.24–4.06)	0.008		ns		
Other solid transplants	1.97 (1.02–3.84)	0.044	2.64 (1.32–5.28)	0.006		
Graft loss censored by death	1.61 (1.03–2.53)	0.038	1.95 (1.23–3.09)	0.005		
GDF-15 medium risk tertile	2.16 (1.14–1.44)	0.018		ns	3.24 (1.2–8.8)	0.021
GDF-15 high risk tertile	3.28 (1.79–6.1)	0.001	2.29 (1.24–4.24)	0.009	4.3 (1.65–11.54)	0.003
EPTS	1.03 (1.02–1.04)	<0.001			1.02 (1.01–1.03)	<0.001

Model 1 included: Age, diabetes, coronary artery disease, peripheral arteriopathy, graft loss censored by death, and GDF-15 tertiles (growth differentiation factor 15). Model 2 included: Estimated Post Transplant Survival (EPTS) score and GDF-15 tertiles. HR: hazards ratios; CI: confidence interval.

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
