# Peer review of "Growth Differentiation Factor 15: A Biomarker with High Clinical Potential in the Evaluation of Kidney Transplant Candidates"

_jcm, 2020, doi:10.3390/jcm9124112_

Round 1
Reviewer 1 Report
The authors performed a retrospective single center analysis of all adult renal transplant recipients from 2005 to 2015 for the value of GDF-15 to predict mortality as primary outcome. Therefore, they analyzed prospectively collected pretransplant serum samples of the patients. They show an independent association of GDF-15 with mortality. Secondly, the predictive value of GDF15 prior to KTx adds accuracy for the EPTS score to predict mortality. The paper is well written and the topic is interesting to the transplant community. The study was performed in a quite big patient cohort of 395 patients with a sufficient mean follow-up of 90.6 months.
However, I have some questions.
- I do not think that Figure 1 is very meaningful. I suppose to dispense with it.
- It hast to be discussed more detailed whether GDF-15 plays a prognostical or also a causal role for mortality and morbidity in patients with ESRD/kidney transplantation and what the potential reasons could be for its prognostical Impact.
- Mortality risk prediction is shown for EPTS score alone and in combination with GDF-15 in Figure 3. Could you also provide ROC-curves for sole GDF-15 without EPTS score?
Author Response
In response to your comments on our article, we would like to thank you for your feedback.
- As it was suggested, we have deleted Figure 1 due to its lack of contribution to the article.
- We have included one paragraph (lines 241-253) in which we discuss the possible sources of GDF-15 as well as its prognostical or causal role for mortality. To the best of our knowledge, we can not answer this question with certainty, and we acknowledge this limitation to translate the use of this marker into practice.
- We agree with the reviewer’s assessment and we have included ROC-curves for GDF-15 tertiles, EPTS and the combination of both in the mortality risk prediction model at 1, 3 and 5 years (lines 193-200). Due to the limitations to compare the three curves (the analysis if performed in pairs) we have specified the comparison between EPTS and GDF-15, and between EPTS and GDF-15 + EPTS respectively. To perform this analysis we used a different software (both we had previously used did not allow this function) so we have included it in the Experimental Section (line 116-120).
Additionally, a native English speaker has reviewed and corrected the manuscript addressing the language issues encountered in it. We hope the revised version will be adequate, but are glad to consider further revisions, and would like to thank you again for your interest in our research.
Sincerely.
Reviewer 2 Report
The role of biomarkers in predicting survival outcomes after kidney transplantation is relevant and could lead towards differential approach in selected recipients. Could the authors also explain what could be the influential tole of donor type, such as DBD, DCD or live donors as well as tailored immunosuppression? I believe the manuscript would benefit form an analysis of tehse factors, too.
Author Response
In response to your comments on our article, we would like to thank you for your feedback.
- As it was suggested, we have added information of donor type. We have included it in the description of results (lines 134-136) and the survival analysis (line 164). Given the limited number of living donor and DCD donors, we have mentioned this as a limitation of the study related to its generalization (line 277).
- Regarding the immunosuppression tailoring, we agree with the reviewer’s assessment and consider that it could be beneficial to consider this factor in our analysis. However, our study aims to focus on the prognostic ability of GDF-15 prior the transplant, when the immunosuppression regimen is not known yet, and every patient had a different adjustment subject to clinical considerations, making the adequate description of this factor (apart from the initial immunosuppression, that was included and analyzed in the article) extremely complex. Additionally, in our cohort GDF-15 related to age and time on RRT, and not other clinical conditions that are usually taken into consideration to change the immunosuppression tailoring, making difficult to influence the results of our study.
We hope the revised version will be adequate in spite of its limitations, but are glad to consider further revisions, and would like to thank you again for your interest in our research.
Sincerely.
Reviewer 3 Report
Comment To Authors
The manuscript is very interesting, also consdering the high cardiovascular morbidity and mortality in kidney transplanted patients.
The manuscript is well written
The sample seem adequate
Growth Differentiation Factor – 15 was already studied from many authors, also in CKD and AKI but effectively has never been analyzed in such a large population of kidney transplant patients.
-Matthias Heringlake, Efstratios I. Charitos, Kira Erber, Astrid Ellen Berggreen, Hermann Heinze, Hauke Paarmann. Preoperative plasma growth-differentiation factor-15 for prediction of acute kidney injury in patients undergoing cardiac surgery. Crit Care. 2016; 20: 317.
.Viji Nair, Cassianne Robinson-Cohen, Michelle R. Smith, Keith A. Bellovich, Zeenat Yousuf Bhat, Maria Bobadilla, Frank Brosius, Ian H. de Boer, Laurent Essioux, Ivan Formentini, Crystal A. Gadegbeku, Debbie Gipson, Jennifer Hawkins, Jonathan Himmelfarb, Bryan Kestenbaum, Matthias Kretzler, Maria Chiara Magnone, Kalyani Perumal, Susan Steigerwalt, Wenjun Ju, Nisha Bansal. Growth Differentiation Factor–15 and Risk of CKD Progression. J Am Soc Nephrol. 2017 Jul; 28(7): 2233–2240.
The cardiovascular risk associated with growth differentiation factor 15 has also already been extensively studied but not in kidney transplant patients
-Shanhui Xie, Liping Lu, Liwei Liu. Growth differentiation factor‐15 and the risk of cardiovascular diseases and all‐cause mortality: A meta‐analysis of prospective studies. Clin Cardiol. 2019 May; 42(5): 513–523.
-Daniel Lindholm, Emil Hagström, Stefan K. James, Richard C. Becker, Christopher P. Cannon, Anders Himmelmann, Hugo A. Katus, Gerald Maurer, José Luis López‐Sendón, Philippe Gabriel Steg, Robert F. Storey, Agneta Siegbahn, Lars Wallentin. Growth Differentiation Factor 15 at 1 Month After an Acute Coronary Syndrome Is Associated With Increased Risk of Major Bleeding. J Am Heart Assoc. 2017 Apr; 6(4): e005580.
Is growth differentiation factor 15 a simple investigation? what are the costs? Can it be used in clinical practice or for research purposes only? Could the authors explain it better?
It would be interesting to know other cardiovascular risk factors especially in this population, such as inflammation indices, mineral metabolism, uricemia.
It would be interesting to know if there is an association with these factors as well.
Furthermore, given that transplant patients are generally very studied, it could be interesting to evaluate a possible association with indices of early systemic atherosclerosis such as IMT or ABI
Author Response
In response to your comments on our article, we would like to thank you for your feedback.
- In response to the first question, GDF-15 has been measured for research purposes by manual methods, in most of the cases using commercially available ELISA such as the one mentioned in our study (Quantikine, R&D Systems, up-to-date price 617.00 €/kit for 96 determinations). However, due to the growing number of studies and potential uses of the molecule, an automated electrochemiluminescence immunoassay has been presented recently, and hopefully it will increase the availability of the test and reduce its price. We have included this information in the article (lines 266-269).
- As it was suggested, we have added information of other stablished cardiovascular risk factors such as inflammation (CPR), uricemia and mineral metabolism. None of this values correlated to GDF-15 or survival in our analysis, but we agree with the reviewer that are of interest and describe the cohort more effectively. We have included them in the description of results (table 1), GDF-15 determinants (lines 146-147) and the survival analysis (line 163).
- We agree with the reviewer that would be of great interest to evaluate the association of GDF-15 with indexes of atherosclerosis such as IMT or ABI. However, they were not systematically studied in our cohort of patients and are therefore not available to be included in our study. We have specified this limitation in the paper (lines 278-281).
We hope the revised version will be adequate, but are glad to consider further revisions, and we thank you for your interest in our research.
Sincerely.
Reviewer 4 Report
The topic is original. The study is well designed and performed. I have no objections to the material and methods. The results are correctly discussed, and the conclusions are justified. The manuscript is well written; however, a few language corrections should be done, that is why I suggested a careful language edition, preferably by an English language native speaker. In my opinion, the paper has a significant citation potential. Summarizing, I expect that the authors address only a few comments.
1. Page 1, line 39: FGF should be replaced with TGF.
2. Careful language edition should be done.
Author Response
In response to your comments on our article, we would like to thank you for your feedback. We have corrected the error located in line 39 (highlighted in yellow). Additionally, a native English speaker has reviewed and corrected the manuscript addressing the language issues encountered in it. We hope the revised version will be adequate, but are glad to consider further revisions, and we thank you for your interest in our research.
Sincerely.
Round 2
Reviewer 1 Report
De Cos Gomez et al. provided a revised version of there manuscript "Growth differentiation factor 15: a biomarker with high clinical potential in the evaluation of kidney transplant candidates". They addressed all of my concerns/comments sufficiently.
Therefore, I recommend this manuscript for publication.
Author Response
Thank you very much for your cooperation and your interest in our research.
Sincerely.
Reviewer 3 Report
the authors reported the changes required, therefore the manuscript can be accepted in its current form
Author Response

(The authors gave the same response as above.)
